# EgoQuestions: Crafting Egocentric Questions for Egocentric Video Question Answering

## Abstract

A thorough understanding of models' egocentric capabilities is crucial for robotics, autonomous driving, smart glasses, etc. Egocentric VideoQA aims to assess models' understanding of first-person videos, but existing benchmarks often include questions that do not reliably probe recorder-centric reasoning. Using these datasets to train and evaluate models can obscure true model capabilities and reduce the value of curated egocentric data. To address this, we define egocentric questions and propose three clear principles: a question should focus on the video recorder and their activities; it must avoid shortcut cues that allow answers via generic scene or action recognition (e.g., simultaneously naming an action and its object); while intentions and attributes may serve as shortcuts for actions and objects, those that require understanding of the recorder's perspective will not. Guided by these principles, we build a checking pipeline to filter existing QA pairs and a crafting pipeline to generate valid egocentric questions. We release EgoQuestions, a benchmark of 2,500 curated egocentric QA instances created with our pipeline, and evaluate several proprietary and open-source VLMs. Results reveal substantial room for improvement in current models' egocentric capabilities and a clear performance gap (about **10**%) between egocentric questions that adhere to our principles and flawed alternatives, demonstrating existing egocentric benchmarks tend to overrate models' first-person capabilities. and the need for rigorously designed egocentric benchmarks to more accurately assess models' first-person vision capabilities.

## 1 Introduction

Egocentric videos refer to recordings captured from a first-person point of view, which have become popular with the advancements of wearable technology (Hodges et al., 2006; Engel et al., 2023). The field of egocentric vision has gained significant attention from researchers in recent years, with applications across various domains, including augmented reality (AR) (Plizzari et al., 2024; Lv et al., 2024), embodied AI (Mu et al., 2023), and human-object interaction (Liu et al., 2022; Wang et al., 2023) among others. To support the development of models for understanding and describing these videos, numerous datasets for egocentric video analysis have emerged (Damen et al., 2018; Grauman et al., 2022; Zhu et al., 2023; Pan et al., 2023; Grauman et al., 2024; Bi et al., 2024; Perrett et al., 2025; Chen et al., 2025), including datasets such as Ego4D and Ego-Exo4D that provide extensive collections of videos. Egocentric video understanding presents additional challenges compared to videos captured from a third-person (exocentric) perspective. The scene of egocentric videos changes constantly and unpredictably, and usually only the recorder's some body parts are visible. More importantly, for effective real-world deployment, a model must be able to perceive its environment from a first-person viewpoint. However, Vision-Language Models (VLMs) trained predominantly on third-person visual data may not possess this essential egocentric capability.

A variety of benchmarks have been proposed to evaluate models' egocentric capabilities, including those that adapt Video Question Answering (VideoQA) to egocentric videos (Fan, 2019; Jia et al., 2022; Bärmann & Waibel, 2022; Mangalam et al., 2023; Cheng et al., 2024b; Di & Xie, 2024; Cheng et al., 2024a; Zhou et al., 2025a; Chen et al., 2025). While these benchmarks encompass egocentric videos and specific tasks, they fail to align the videos with corresponding genuinely egocentric questions, i.e., the questions regarding the video recorder's intention, action, and perception of the

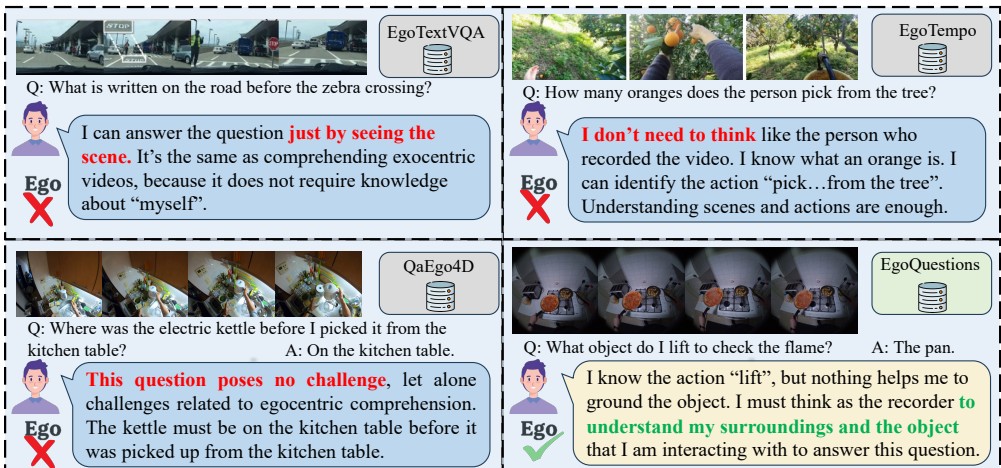

Figure 1: Instances in EgoTextVQA (Zhou et al., 2025a), EgoTempo (Plizzari et al., 2025), QaEgo4D (Bärmann & Waibel, 2022) and EgoQuestions. We identify shortcomings in the questions contained in the three existing benchmarks and describe them respectively in the image above.

scene. For instance, as shown in Figure 1, some questions (top-left) focus on general scene content, which is unrelated to the recorder's perspective. Others (top-right) allow models to rely on non-egocentric cues like spatial grounding or simple action recognition, bypassing the need for genuine egocentric reasoning. Furthermore, some questions are poorly constructed, such as those where the answer is implied within the question itself (bottom-left). These question flaws may misleadingly suggest that current models have a strong egocentric understanding while diminishing the value of curated egocentric video data. We believe that evaluating a model's first-person capabilities requires not only first-person perspective videos but also carefully crafted **questions** tailored to these scenes.

To address defects in existing questions, in this paper, we aim to craft **egocentric questions** tailored for **egocentric tasks**. We first define egocentric questions and state three principles they must follow. We propose that the question must focus on the camera wearer, not merely on the environment. Also, the question must avoid shortcuts that let models rely solely on general scene or action recognition—for example, it should not simultaneously name the action and the object that makes the answer obvious. Finally, the question may ask about the wearer's intentions or attributes when those aspects require understanding the recorder's role. Based on these deterministic principles, we develop a checking pipeline to evaluate existing question–answer pairs and a crafting pipeline to generate valid egocentric questions.

We introduce the EgoQuestions benchmark, which includes 2,500 instances of egocentric questions paired with curated egocentric video clips, all generated using our crafting pipeline. We evaluate several recent VLMs, including proprietary and open-source models. Results show substantial room for improvement in current VLMs' egocentric understanding. To illustrate the effect of flawed questions, we also construct corresponding question sets that violate our principles; performance on these flawed questions is substantially different from performance on our egocentric questions, underscoring the need to use proper egocentric questions in egocentric VideoQA.

Our contributions can be summarized in three key abstracts.

**We identify current shortcomings in questions and propose a set of principles for egocentric questions.** Ours is the first work to identify defects in questions in current egocentric VideoQA benchmarks. Furthermore, we propose explicit requirements that egocentric questions must satisfy and provide principles for validating them.

**We develop a benchmark, EgoQuestions, that encompasses crafted egocentric questions.** We curate the questions of EgoQuestions with our question-crafting pipeline, producing questions of substantially higher quality than those in current benchmarks. We further employ this benchmark to evaluate recent VLMs, reporting results that reflect the models' egocentric capabilities. We also

demonstrate the performance gap caused by problematic questions by constructing a comparison set of flawed questions, which indicates that existing egocentric benchmarks tend to overestimate models' first-person capabilities.

**We implement a crafting pipeline to generate egocentric questions.** We will release our full crafting pipeline for egocentric question generation, enabling the community to generate more egocentric questions and facilitate the development of VLMs.

## 2 RELATED WORK

### 2.1 EGOCENTRIC VIDEO QUESTION ANSWERING BENCHMARKS

Video question answering is a task that requires a joint understanding of both visual content and the corresponding textual information. With the growing accessibility of wearable photographic devices and the emergence of large-scale egocentric video datasets (Grauman et al., 2022; Wang et al., 2023; Grauman et al., 2024; Perrett et al., 2025), previous studies have investigated the integration of textual cues with egocentric videos, leading to the development of egocentric VideoQA benchmarks. Among these, EgoTempo (Plizzari et al., 2025) emphasizes temporal comprehension of videos, whereas EgoTextVQA (Zhou et al., 2025a) is built to evaluate the scene-text understanding capabilities of models. Additional addressed challenges include long-form video orientation (Mangalam et al., 2023; Zhou et al., 2025b), goal understanding (Jia et al., 2022), social norm interpretation (Rezaei et al., 2025), multi-hop video question answer-

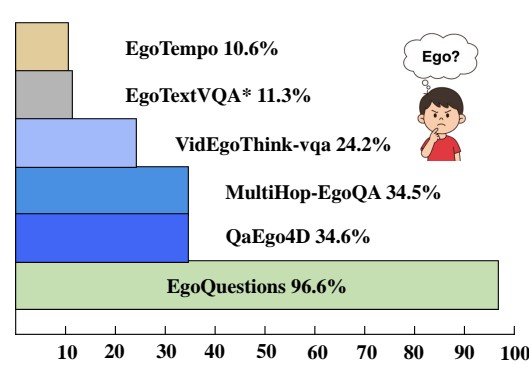

Figure 2: A visualization of the percentage of egocentric questions contained in current egocentric VideoQA benchmarks and EgoQuestions. * represents that we only evaluate a subset of EgoTextVQA.

ing (Chen et al., 2025), cross-domain generalization (Li et al., 2025), *etc*.

While defining tasks to utilize egocentric videos, prior works have largely overlooked the linguistic formulation of the corresponding questions. The percentage of egocentric questions contained in current benchmarks are depicted in Figure 2. This may lead to emergence of inferential shortcuts that models can identify, resulting in a gap between evaluation outcomes and the model's true egocentric capabilities. In our work, we introduce a definition of egocentric questions accompanied by a framework for generating such questions from narrative descriptions, thereby mitigating this possible discrepancy. As shown in the figure, while other benchmarks contain less than 40% instances with egocentric questions, EgoQuestions boasts a percentage of over 95%.

### 2.2 ANNOTATIONS METHODS INVOLVING LLMS / MLLMS

The recent development of large language models (LLMs) and multi-modal large language models (MLLMs) (Brown et al., 2020; Ouyang et al., 2022; Touvron et al., 2023; Team et al., 2023; Grattafiori et al., 2024; OpenAI et al., 2024; OpenAI, 2024; DeepSeek-AI et al., 2025) has enabled the usage of these models for benchmark construction. Among benchmarks concerning Egocentric VideoQA, several prior studies have employed LLMs / MLLMs in their data generation pipelines, primarily leveraging the extensive annotations available in large-scale egocentric video datasets such as Ego4D (Grauman et al., 2022). EgoSchema (Mangalam et al., 2023) utilizes LLMs in both the QA generation and filtering stages, while EgoTextVQA (Zhou et al., 2025a) and VidEgoThink (Cheng et al., 2024a) integrate the usage of the GPT-4o model (OpenAI, 2024) in their QA generation processes.

However, even when provided with detailed prompts, LLMs and MLLMs often struggle to meet complex requirements, necessitating human filtering after LLM / MLLM usage. In our work, we

utilize code templates to systematically generate questions and answers, leveraging LLMs exclusively for grammar component extraction and question refinement. This methodology enables more fine-grained control over the format of questions.

# 3    WHAT MAKES A GOOD EGOCENTRIC QUESTION

In this section, we introduce the defects present in question formulations of current egocentric benchmarks. Subsequently, we propose three guiding principles for the design of egocentric questions and provide an evaluation of existing egocentric VideoQA benchmarks in light of these principles.

## 3.1    DEFECTS IN CURRENT QUESTIONS

The objective of advancing model comprehension of egocentric videos is to facilitate model development in real-world applications, such as personalized agents and embodied AI. Therefore, egocentric VideoQA benchmarks should be designed to assess models' capacity for understanding surrounding environments and interactions from a first-person perspective. However, an examination of current egocentric VideoQA benchmarks reveal that many questions fail to meet this objective, either by not addressing the subject at all or by offering inference shortcuts that do not require egocentric understanding. For instance, the question "What is the man in red clothes doing?" does not concern the video recorder (*i.e.* myself), making it equally applicable to third-person VideoQA tasks. More subtle flaws arise in questions like "What am I cutting that is on the table?". A model proficient in scene identification and action recognition could answer this by detecting the action ("cut") and object location ("on the table") without egocentric comprehension. These deficiencies may result in a misalignment between a model's egocentric capabilities and its performance on related benchmarks.

## 3.2    THREE PRINCIPLES OF EGOCENTRIC QUESTIONS

To systematically assess existing questions and create new egocentric questions, we propose three principles that such questions should follow.

**Principle 1: The question must concern the recorder.** To ensure models adopt the recorder's perspective, questions must pertain to the recorder themselves. A question unrelated to the recorder could be equally answered using third-person (exocentric) videos, reducing egocentric videos to conventional exocentric videos with frequent camera movement. For example, "What am I doing?" is an egocentric question, while "What is the man in red clothes doing?" is not.

**Principle 2: Avoid shortcuts related to both the action and the object.** With the subject restricted to the recorder by Principle 1, there must not be shortcuts related to both the action and the object being interacted with. Otherwise, an MLLM capable of interpreting actions and scenes could answer the question without egocentric understanding. For objects, stating the object itself or its attributes (independent of the recorder) can create shortcuts. A similar logic applies to actions and their associated adverbs. For example, "What am I cutting?" is an egocentric question, but "What am I cutting that is on the table?" is not. Further details are discussed in the following principle.

**Principle 3: Attributes or intentions relevant to the recorder are allowed.** This principle complements Principle 2. While some attributes may introduce inference shortcuts, those directly relevant to the recorder are unlikely to do so. The model must still understand the recorder's role to comprehend them. The same reasoning applies to the intentions of the actions.

We observe that the aforementioned principles can be reformulated as rules concerning the grammatical components of both simple and complex sentences. Specifically, assuming the question and answer can be restructured into a simple or compound sentence, Principle 1 requires the subject to be the recorder; Principle 2 concerns the appearance of the predicate, object, attributives, and adverbials in the question; and Principle 3 focuses on attributives and adverbials. The relationship helps determine whether a question is truly egocentric.

## 3.3    REVIEWING CURRENT BENCHMARKS

Building upon the above three principles, we implement a framework that utilizes LLMs to identify actions, objects, and other components in questions and answers, and to determine whether

Table 1: Evaluation results of questions in QaEgo4D (Bärmann & Waibel, 2022), the VQA task in VidEgoThink (Cheng et al., 2024a), EgoTextVQA (Zhou et al., 2025a), EgoTempo (Plizzari et al., 2025), and MultiHop-EgoQA (Chen et al., 2025) with our framework. * indicates that we only evaluate a certain subset of the items, with details contained in the appendix.

| Benchmark | Video Source | Q-A Source | Percentage(%) |
|---|---|---|---|
| QaEgo4D | Ego4D (Grauman et al., 2022) | Ego4D NLQ | 34.59 |
| VidEgoThink-vqa | Ego4D | Ego4D Narrations | 24.17 |
| EgoTextVQA* | RoadTextVQA, EgoSchema | Videos | 11.34 |
| EgoTempo | Ego4D | Ego4D Narrations | 10.6 |
| MultiHop-EgoQA | Ego4D | Ego4D Narrations | 34.54 |

a question conforms to our principles. We apply this framework to five benchmarks containing open-ended egocentric VideoQA instances; the percentages of the questions adhering to the three principles, along with each benchmark's video and QA sources, are shown in Table 1. We exclude EgoSchema(Mangalam et al., 2023) and the HD-Epic VQA benchmark(Perrett et al., 2025) in this evaluation because their multiple-choice instances can depend heavily on the provided options. For example, an EgoSchema item asks the model to "briefly describe this interruption and its significance within the video," with detailed descriptions given among the choices. The results show that many existing works overlook the importance of well-structured questions and do not follow our proposed principles. Improving question formatting and adherence to these principles could increase the quality and reliability of responses in VideoQA tasks.

## 4 EGOQUESTIONS DATASET

In this section, we provide an overview of the construction process of EgoQuestions. We introduce the video collection methods and describe the pipeline used for generating question-answer pairs.

### 4.1 VIDEO COLLECTION

Our video data are sourced from two recognized and publicly accessible egocentric video datasets: Ego4D(Grauman et al., 2022) and HD-Epic(Perrett et al., 2025). Both datasets contain a diverse variety of manually collected videos. Ego4D includes more than 3,670 hours of egocentric videos, recorded in 74 locations across 9 different countries. HD-Epic's annotators recorded their kitchen activities over at least three days, producing egocentric videos that capture a wide array of objects and actions within complex indoor environments.

The natural language queries (NLQ) task, part of the Episodic Memory benchmark of Ego4D, provides samples with a natural language query $Q$ and a time slot $r$ that serves as the ground truth for when the answer to $Q$ occurs. QaEgo4D (Bärmann & Waibel, 2022) was built upon NLQ annotations, employing human annotators to curate answers to $Q$. These responses form complete items consisting of a video $V$, a question $Q$, a time slot $\tau$, and a newly curated answer $A$. In our project, we directly use the $V$ and $\tau$ from QaEgo4D items for EgoQuestions. Additionally, HD-Epic's diverse video content provides broad coverage of indoor activities, which complements the international and varied settings presented in Ego4D, enhancing the robustness of our dataset.

The second method of utilizing Ego4D videos is based on the dataset's original narrations, each of which includes a timestamp and a brief description of the camera wearer's activities at that moment. Previous studies (Lin et al., 2022; Plizzari et al., 2025) have constructed time slots based on these narrations. We detail their methodologies and present our analysis in the appendix. In our approach, we first sort all the narrations for a video according to their timestamps. We then define the time slots using the following strategy:

$$\tau_k = (\text{random}(t_{k-1}, t_k), \quad \text{random}(t_k, t_{k+1})), \tag{1}$$

where $\tau_k$ represents the constructed time slot for the $k$th narration, and $t_k$ is the original timestamp of that narration. By employing this approach, we account for the density of narrations; in particular, the duration of each time slot is extended based on the spacing between narrations. This helps mitigate the influence of isolated narrations and reduces distractions caused by adjacent narration timestamps within our time intervals.

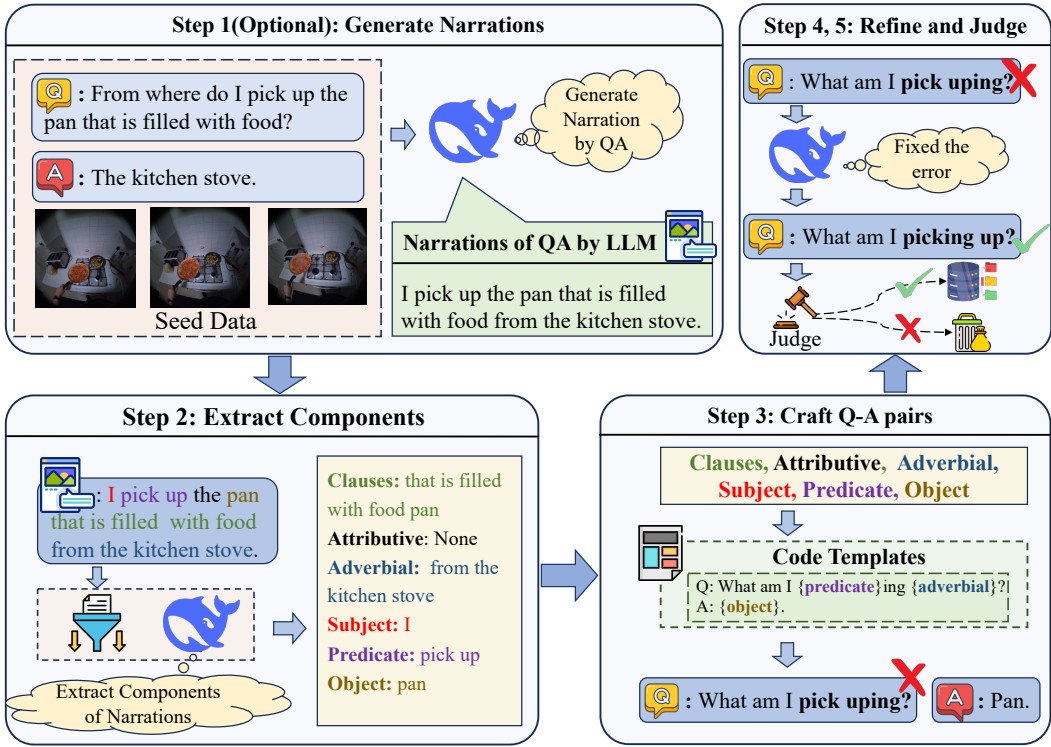

Figure 3: Overview of our egocentric QA-generation pipeline. The five image panels correspond to the five steps described in Sec. 4.2. Soft yellow rectangles (steps 1 and 4) indicate LLM-based procedures; green rectangles indicate purely code-based processes (step 3). Gradient rectangles (steps 2 and 5) denote procedures that combine LLM usage and code processing.

For videos in the HD-Epic dataset, we employ the video id and time slots provided by the vision inputs in the HD-Epic VQA benchmark directly.

## 4.2 QA GENERATION

Following the description of our method for acquiring egocentric videos and constructing time slots, we now present a detailed overview of our pipeline for generating egocentric questions and their corresponding answers. An illustrated overview of this pipeline is provided in Figure 3.

**(Optional) Step 1: Generating narrations from question-answer pairs.** This initial step is exclusively applied when utilizing the question-answer pairs derived from the QaEgo4D dataset (Bärmann & Waibel, 2022). For Ego4D (Grauman et al., 2022) and HD-Epic (Perrett et al., 2025), which both provide narrations directly, there is no need for Step 1. To generate narrations for subsequent steps, we prompt a capable large language model (LLM) to produce an objective, declarative sentence based on a given question-answer pair. For example, if the question is "What did I put in the refrigerator?" and the answer is "A pack of raisins," the generated sentence should be "I put a pack of raisins in the refrigerator." We observe that the employed LLM demonstrates high proficiency in performing this transliteration task, as detailed in the appendix.

**Step 2: Extracting grammatical components from narrations.** Using video narrations, we apply a multi-step extraction process with multiple LLM calls to identify each sentence's grammatical components. Accurate execution of this process is crucial for the subsequent question generation step, as it requires precise control over the grammatical elements. Our investigation reveals that even an advanced LLM such as Deepseek-V3.1 is unable to accurately extract all grammatical components in a single pass. To address this, we designed a chained extraction process that first extracts clauses, followed by attributives and adverbials, and finally identifies the subject, predicate, and ob-

ject. We find that the LLM can extract only a limited number of components per call when previously extracted components are removed, with detailed results provided in the appendix.

For instances where the original narration consists of multiple sentences or includes a compound sentence, we use an LLM to classify its type beforehand, retaining only the longest sentence or segment for component extraction. In cases where the narration (or preprocessed content) contains grammatical errors, these issues often become evident after the extraction process, manifesting as incomplete components—for example, the absence of a predicate.

**Step 3: Crafting egocentric questions with code templates.** Building on the extracted grammatical components, we proceed to construct egocentric questions following established principles. Previous studies (Mangalam et al., 2023; Cheng et al., 2024a; Zhou et al., 2025a; Chen et al., 2025) have utilized LLMs or MLLMs to generate question-answer pairs, often providing guidance and examples in their prompts to facilitate this process. These studies typically apply human-based or LLM-based post-generation filtering. Although this combination of generation and filtering can substantially improve the overall quality of questions, the outputs remain heavily influenced by the examples included in the prompts, limiting the control needed to produce truly egocentric questions. To address this, we define code templates that ensure the structural integrity of the questions, completing their content with the relevant components. We establish a minimum of five templates for each question format to enhance diversity in the generated questions.

In total, three types of questions are generated, all adhering to our principles. The first type of question includes the subject and predicate, asking about the object of the narration. The second type provides the subject, object, and its corresponding attributive, while inquiring about the predicate. The third type extends the first type by incorporating adverbials from the original sentence. Our approach maintains a high degree of control by structuring questions using predefined templates. Additionally, since grammar extraction is a process with a ground truth, our method helps to reduce errors that are difficult to detect when using MLLMs to generate questions from visual context.

**Step 4: Refining questions via LLM.** While combining components with code templates ensures the grammatical structure of the question, it can sometimes result in issues such as a lack of fluency or incorrect spelling. To address this, we use an LLM to refine the question while preserving its original meaning whenever possible. For example, if the generated question is "What of dough object is involved when I Add gently?", the template does not handle the attributive or capitalization properly. The LLM corrects these errors, refining the question to "What dough object is involved when I add gently?" without altering its overall meaning. We observe that the LLM can fix most issues related to fluency.

**Step 5: Post-generation filtering.** Previously, we implemented code to evaluate questions from existing egocentric VideoQA benchmarks. To ensure that our generated questions meet our specific requirements, we use this code to filter our own questions, removing those deemed invalid by the program. Additionally, we perform supplementary filtering steps, such as eliminating instances where the answers are contained within the questions.

## 4.3 DATASET ANALYSIS

EgoQuestions comprises 2,500 VideoQA instances, procured from the previously introduced video collection and QA generation processes. Our videos are sourced from two primary sources: Ego4D (Grauman et al., 2022) and HD-Epic (Perrett et al., 2025), sampled from a total of 260 raw videos. We categorize the data into three subsets based on the three types of questions previously described in Step 3 of the QA generation process. Dubbed as SPO (because we provide the Subject and Predicate in the question while inquiring about the Object), SOAP, and SPAO, the three subsets contain 1,146, 695, and 659 instances, respectively. We present the average duration of the video clips, the average word count of our questions, and other relevant information in Table 2.

## 5 EXPERIMENTS

In this section, we evaluate both API-based and open-source vision–language models on the Ego-Questions benchmark to demonstrate the importance of using egocentric questions. We use these results to answer the following two questions:

Table 2: Dataset stats. We display the number of instances, the number of raw videos, the average clip duration, the average word count in our questions and the source of our questions for each of our subsets in this table. In the Q-A Source column, X+Y+Z denotes X questions are constructed from HD-Epic annotations, Y questions are constructed from Ego4D narrations, and Z questions are constructed from QaEgo4D Q-A pairs.

| Subset | Instances | Raw Videos | Q-A Source | Avg. clip duration | Avg. Q word count |
|--------|-----------|------------|------------|--------------------|--------------------|
| SPO    | 1146      | 207        | 221+738+187 | 3.87              | 5.88              |
| SOAP   | 695       | 160        | 167+475+53  | 3.99              | 9.48              |
| SPAO   | 659       | 151        | 152+467+40  | 3.51              | 9.88              |

**Question 1:** How well can current models answer egocentric questions about egocentric videos?

**Question 2:** To what extent do egocentric questions affect experimental outcomes compared with standard exocentric questions?

## 5.1 EVALUATION

Following prior work on open-ended question answering evaluation (Cheng et al., 2024b;a), we use a capable LLM as a judge, presenting it with the question, the ground truth, and the model's response. The judge scores each answer as 0, 0.5, or 1, where 0.5 indicates a partially correct response. We provide more details, such as our evaluation prompt, in the appendix.

## 5.2 MODELS

Using EgoQuestions, we evaluate the egocentric capabilities of several recent vision–language models, including 3 API-based models and 5 open-source models. For API-based models, we evaluate GPT-5 (Seed, 2025a), doubao-seed-1.6-flash and doubao-seed-1.6(Seed, 2025b). For open-source models, we evaluate Qwen2.5-VL-7B-Instruct(Bai et al., 2025), Qwen2-VL-7B-Instruct (Wang et al., 2024), GLM-4.1V(Team et al., 2025), MiniCPM-V-4.5(Yao et al., 2025), and InternVL-3.5-8B (Wang et al., 2025), encompassing both widely recognized models such as Qwen-2.5-VL-7B-Instruct and the latest models such as MiniCPM-V-4.5 and InternVL-3.5-8B (Both of which were released in August 2025).

## 5.3 GENERATION OF INVALID COUNTERPARTS

The question–answer pairs in EgoQuestions can effectively evaluate VLM capabilities and therefore address Question 1. However, Question 2—comparing egocentric and exocentric questions—requires further development. EgoVQA (Fan, 2019) used a similar approach to measure the performance gap between first-person and third-person questions. Since the repository linked in the original paper is no longer accessible, we examine the paper's examples and descriptions, which suggest the comparison may be unfair: first-person and third-person questions often target different aspects of the videos, making it difficult to isolate and mitigate the influence of other factors on model performance. Moreover, some Egocentric VideoQA examples in the paper contain questions that do not conform to our principles.

We argue that a rigorous comparison between egocentric and exocentric questions must meet the following requirements. First, both question types should refer to content within the same video clip. Second, paired questions for the same video should elicit identical answers and differ only in phrasing or structure. Guided by these requirements, we create exocentric question templates for comparison. Although these templates differ only slightly from the original egocentric templates, they can produce questions that provide shortcuts, allowing models to answer correctly without possessing egocentric capabilities. Apart from the template modifications, the generation process for the comparison questions is identical to that used for the original egocentric questions.

To ensure a fair comparison, we replace the code templates used in Step 3 of the QA generation pipeline with corresponding templates that produce similar but exocentric questions. This yields question pairs that share the same answer and video clip. We denote the egocentric questions as T1 (T for True) and their exocentric counterparts as F2 (F for False).

Table 3: Experimental results on the three subsets of EgoQuestions. T1 denotes the originally constructed egocentric questions, while F2 means the corresponding non-egocentric questions specifically crafted to ensure a rigorous comparison. For a subset, $\Delta_{acc}$ represents the accuracy gap between the F2 and the T1 accuracy, reflecting the impact of non-egocentric question usage.

| Model | SPO-Comparison | | | SOAP-Comparison | | | SPAO-Comparison | | |
|---|---|---|---|---|---|---|---|---|---|
| | T1 | F2 | $\Delta_{acc}$ | T1 | F2 | $\Delta_{acc}$ | T1 | F2 | $\Delta_{acc}$ |
| GPT-5 | 34.6 | 45.3 | 10.7 | 24.7 | 37.2 | 12.5 | 40.5 | 53.6 | 13.1 |
| Doubao-1.6 | 32.8 | 45.3 | 12.5 | 24.1 | 34.7 | 10.6 | 36.7 | 46.8 | 10.1 |
| Doubao-1.6 flash | 30.1 | 41.6 | 11.5 | 19.5 | 27.9 | 8.4 | 32.9 | 44.0 | 11.1 |
| Qwen2.5-VL-7B-Instruct | 29.5 | 39.4 | 9.9 | 21.9 | 31.1 | 9.2 | 32.3 | 40.4 | 8.1 |
| Qwen2-VL-7B-Instruct | 30.0 | 38.0 | 8.0 | 18.3 | 27.8 | 9.5 | 34.0 | 38.8 | 4.8 |
| InternVL3.5-8B | 26.4 | 38.6 | 12.2 | 22.9 | 31.5 | 8.6 | 30.4 | 41.6 | 11.2 |
| GLM-4.1V-Thinking | 25.8 | 35.4 | 9.6 | 15.2 | 22.0 | 6.8 | 29.1 | 39.1 | 10.0 |
| MiniCPM-V 4.5 | 26.1 | 36.4 | 10.3 | 19.8 | 32.4 | 12.6 | 31.6 | 37.8 | 6.2 |

## 5.4 RESULTS

After evaluating open-source and API-based VLMs, we provide answers to the two questions that we posed above. For Question 1, current VLMs show modest performance on our open-ended egocentric questions. As shown in the T1 columns, the accuracies of VLMs range from 15% to over 40% across subsets. GPT-5 is dominant across all three subsets, outperforming the second-place model by more than 3% on the SPAO subset, while Doubao-1.6 firmly secures the position of the runner-up. Among the five open-source VLMs, no single model stands out. We can also observe the same trend across subsets for all the 8 models. All models perform best on the SPAO subset and perform the worst on the SOAP subset, proving that VLMs are still unable to grasp actions as well as they can ground objects.

For Question 2, we visualize the performance gap between ill-formatted questions and carefully crafted egocentric questions. As shown in the $\Delta_{acc}$ columns, all evaluated API-based and open-sourced VLMs exhibit a significant increase in accuracy when answering exocentric questions instead of the corresponding egocentric instances, proving that a simple shift in the format of questions can indeed result in the overrating of models' performance, leading to misjudgements concerning the egocentric abilities of a model. Take the subject SPO for example. When evaluated on the corresponding flawed questions, GLM-4.1V-Thinking, which scored lowest on the egocentric questions in SPO, scored a percentage of 35.4%, exceeding the accuracy of GPT-5 on egocentric questions. An overrating percentage of more than 10% can shadow even the gap between the abilities of the model itself and GPT-5, which is among the most powerful models nowadays.

Moreover, the tendency of overrating not only appears with models that are less advanced and consist of less parameters. The performance gap appears regardless of a model's absolute performances: although GPT-5 achieves the highest accuracy on all six question formats (three egocentric and three exocentric), it still shows a $\Delta_{acc}$ of over 10% for all three subsets. The results and discussion above further demonstrate the importance of using egocentric questions in egocentric VideoQA.

## 6 CONCLUSION

In this work, we examined prior egocentric VideoQA benchmarks and identified shortcomings in their questions. We proposed three principles for egocentric questions and developed a QA-generation pipeline based on these principles to produce egocentric questions and corresponding answers. Our benchmark, EgoQuestions, comprises Q–A pairs generated by this pipeline. Experiments on EgoQuestions show that using non-egocentric questions can substantially inflate performance, demonstrating the importance of egocentric questions for evaluating models' egocentric capabilities. We will open-source our data and code to promote future research on egocentric VideoQA and related evaluations.

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

## A  LLM USAGE

Throughout the study, we employed LLMs to assist solely in the paper writing process. The models mainly enhanced our original writing by fixing grammatical mistakes and refining the phrasing to clarify the text.

We **did not** employ LLMs to search for relevant works to put into our related works section. We looked up all these works by ourselves.

LLMs **did not** contribute to the intellectual development of the research. The ideas, analyses, and conclusions are all our own.

## B  IMPLEMENTATION DETAILS

We adopt a frame extracting method similar to the method employed in the VQA benchmark of HD-Epic (Perrett et al., 2025). Specifically, we process the videos at 1fps and reformat the frames to a resolution of 336*336. If the number of frames extracted exceeds 8, we uniformly sample the frames to preserve 8 frames. This method is nearly identical to the method in HD-Epic employed when assessing the model VideoLLaMA2, differing only in the upper limit of selected frames (16 for VideoLLaMA2 evaluation). We employ the same set of frames for the evaluation process of all the models, including API-based models such as GPT-5.

## C  DISCUSSION OF TIME SLOT SELECTION FOR EGOQUESTIONS

As mentioned in Section 4.1, previous studies (Lin et al., 2022; Plizzari et al., 2025) have utilized the Ego4D (Grauman et al., 2022) Narrations and generated time slots correspondingly. In the EgoClip pretraining dataset proposed in Lin et al. (2022), the time slot for the ith event is determined as:

$$[t_i^{start}, t_i^{end}] = [t_i - \beta_j/2\alpha, t_i + \beta_j/2\alpha], \qquad (2)$$

in which $t_i$ represents the timestamp corresponding to the ith narration. Suppose the narration is contained in the annotations of the j-th video. $\beta_j$ represents the average time length between adjacent narration timestamps for this video, while $\alpha$ represents the average values of the $\beta$ values among all the videos. We identify an $\alpha$ value of approximately five seconds. As the previous work claims, this format is reasonable because the length of the time slot can be controlled by both the density of identified actions on this video and the overall narration granularity. For example, if a video contains closely-distanced narrations because actions occur constantly in the video, the time slots for narrations on this video will be shorter.

However, from the equation, we expect an average length of 1 s if we generate instances for all narrations. The proposed method also always places the timestamp in the middle; this may be

effective for constructing a pretraining set, but is not required for an evaluation set. Therefore, we develop the approach described in Section 4.1. Our method significantly lengthens the time interval while still accounting for the narration density in each video.

EgoTempo (Plizzari et al., 2025) extends the time length used in EgoClip as well. Because EgoTempo requires longer videos, the authors combine up to 120 narrations to generate video clips. In contrast, our method constructs time intervals that do not overlap with any other narration, reducing contamination from surrounding narrations.

## D   Subset Selection for evaluation in Table 1

In Section 3.3, we evaluate only a certain subset of items from the EgoTextVQA benchmark (Zhou et al., 2025a).

We selected only a subset because the properties of the benchmark suggest a low percentage of egocentric questions. As a dataset designed to evaluate the egocentric scene-text QA ability of models, many questions ignore the recorder and focus solely on scene text. Moreover, EgoTextVQA contains over 7,000 questions; evaluating them all with our validation framework would be costly, as each Q–A instance requires multiple API calls. Therefore, we evaluate only the Gameplay subset of EgoTextVQA-Indoor. We find that more than half of the questions fail to meet our first principle, which requires egocentric questions to concern the recorder.

## E   Validation of LLM-based Steps

In our work, we utilize large language models (LLMs) for certain decidable tasks, such as the extraction of the grammatical components of a sentence. To ensure the trustworthiness of the outputs of these tasks, we conduct human validation of these tasks and provide the results here.

**Generating Narrations from Q-A pairs.** In step 1 of our QA generation pipeline, we employ an LLM to generate a narration from a question and its corresponding answer. Humans can perform this process with ease by fitting the answer in the correct place in the question. For example, we can rewrite the question "What did I put in the refrigerator?" and the answer "pack of raisins" into the objective sentence "I put a pack of raisins in the refrigerator. A correct transliteration should be fluent, preserving the information in the original Q-A pair while not introducing additional information. Upon inspecting 210 instances, we observe an accuracy of about 95.7% (201/210).

**Extracting Grammatical Components.** A given simple or complex sentence contains grammatical components such as a subject, a predicate, objects, attributives, adverbials, and clauses. Students who study grammar in school can often extract these components with high accuracy. However, we find that some advanced LLMs fail when prompted to extract all components in a single call. Therefore, in Step 2 of our QA-generation pipeline, we use three LLM calls, simplifying the sentence after each call. A correct extraction should place every word or phrase in its corresponding place. Upon inspecting 200 instances, we observe an accuracy of 94% (188/200) It is worth noting that the model's strictness toward attributive extraction varies over time. For example, when processing the sentence "I lift the cooking pan.", the model may determine "cooking pan" to be the object now, while determining "pan" as the object and "cooking" as an attributive related to it later. We treated both results above as correct cases during our validation.

## F   Prompt Hubs

### F.1   Evaluation prompt

To evaluate the output of open-ended question answering, we prompt an LLM to generate a score of 0, 0.5 or 1, providing the question, ground truth and the output of the model. We employ the following prompt:

**[Instruction]**

- Please act as an impartial judge and evaluate the quality of the response provided by an AI assistant to the user question displayed below.

- Your evaluation should consider correctness and helpfulness. You should especially **focus on the verbs** when evaluating the response.

- You will be given a reference answer and the assistant's answer.

- The assistant has access to an image alongwith questions but you will not be given images. Therefore, please consider only how the answer is close to the reference answer.

- Be as objective as possible. Discourage uninformative answers.

- Also, equally treat short and long answers and focus on the correctness of answers.

- After providing your explanation, you must rate the response with either 0, 0.5 or 1 by strictly following this format: "`[[rating]]`", for example: "`Rating:   [[0.5]]`".

**[Question]**

*{question}*

**[The Start of Reference Answer]**

*{ground_truth}*

**[The End of Reference Answer]**

**[The Start of Assistant's Answer]**

*{answer}*

**[The End of Assistant's Answer]**

F.2    INFERENCE PROMPTS

For instances constructed from the HD-Epic dataset, we utilize the following prompt wrapping: You are an expert video analyzer, and your job is to answer the open-ended question by giving only a short response. Do not give any other information.

You must give an answer, even if you are not sure.

Question:{question}

Short answer:

During InternVL3.5-8B inference, we utilize the method in the original demo code, placing "Frame-i: <image>
in front for every frame extracted.

For other VLMs, we prompt the model with the generated question directly, paired with the corresponding frames.

