# OpenReview forum: "EgoQuestions: Crafting Egocentric Questions for Egocentric Video Question Answering"
_ICLR.cc/2026/Conference — ICLR 2026 Conference Withdrawn Submission_

### Official Review · Reviewer_prCC · 2025-10-25

**Soundness:** 2
**Presentation:** 3
**Contribution:** 2
**Rating:** 2
**Confidence:** 4

**Summary:**

This paper aims to reflect the challenges of egocentric VideoQA from the question side. It argues that existing egocentric VideoQA datasets are not 100% egocentric, and emphasizes a subset of questions focusing on the camera wearers which cannot be answered with generic visual understanding. The authors design three principles to curate more challenging egocentric questions, and conduct experiments to show that there is a huge gap between existing VLMs in answering generic questions about egocentric visual scenes and answering true egocentric questions.

**Strengths:**

1.	The paper delves into existing egocentric VideoQA datasets and resolves non-egocentric short-cuts, thus exposing the true challenge of answering egocentric questions.
2.	It designs 3 principles to generate true egocentric questions, and constructs a related dataset EgoQuestion.
3.	The analyses show that EgoQuestion is harder to answer than generic questions about egocentric visual scenes.

**Weaknesses:**

1.	While the paper identifies an interesting problem, I do not quite agree that egocentric understanding should be restricted to such narrowly defined questions. The core of egocentric understanding lies in its visual perspective. Therefore, any question grounded in an egocentric visual scene should be considered relevant to egocentric understanding.

2.	The three principles are also problematic. For the first principle (P1), it is hard to justify what questions are related to the recorder themselves. In my opinion, it depends on specific situations and user intentions. For example, all questions seeking for egocentric assistance should be considered pertaining to the video recorders. As for the second principle (P2), it seems more focused on reducing contextual cues to increase task difficulty, rather than reflecting the essence of egocentric understanding. Principle 3 (P3) is rather vague, I feel hard to grasp how it helps to shape the question in EgoQuestion.

3.	Some recent egocentric QA datasets (EgoLife-QA, EgoBlind) that target at egocentric visual assistance have been ignored. According to the 3 principles, I feel hard to be convinced the distinct contribution of EgoQuestion for egocentric understanding compared with these datasets

4.	EgoQuestion’s QAs are automatically generated using LLM and code template. Without human participation, the quality and naturalness of the generated QAs could be questioned. It would be better to attach the annotations or provide some example QAs for understanding. Also, the dataset is relatively small (260 videos and 2500 questions) compared with other egocentric VideoQA datasets.

5.	The experiments and analyses so far are insufficient. The current experiments only demonstrate that the curated questions are more challenging to answer, but do not investigate or suggest directions for solving the challenge. For example, is the performance degradation because of a data shift between model developing and testing? Could crafting similar questions for training improve model performance?

**Questions:**

1.	What do SOAP and SPAO mean?

---

### Official Review · Reviewer_8u9x · 2025-10-28

**Soundness:** 2
**Presentation:** 3
**Contribution:** 2
**Rating:** 2
**Confidence:** 4

**Summary:**

This paper articulates three principles that define what constitutes an egocentric question, presents a semi-automatic pipeline for constructing egocentric QA pairs, and releases EgoQuestions, a curated question set derived from Ego4D and HD-Epic. The authors evaluate a range of off-the-shelf VLMs on this benchmark and demonstrate a consistent performance gap that underscores how question formulation can materially affect assessments of egocentric understanding. However, because the dataset primarily rephrases existing video content rather than adding new data, and no fine-tuning experiments are provided to demonstrate training utility, its contribution remains largely diagnostic and does not clearly advance model capabilities.

**Strengths:**

1. The proposed pipeline is a pragmatic compromise between rule-based control and LLM flexibility. The design is reproducible and the figures/flowcharts in the paper make the procedure clear.
2. By holding video and answer fixed and only varying question wording/structure, the paired design metric (i.e. T1 vs F2) show that question formulation can materially inflate model scores.

**Weaknesses:**

1. Table 3 reports off-the-shelf model inference performance on two question formats (T1 vs F2). This demonstrates that evaluation is sensitive to question phrasing, but it does not establish that EgoQuestions as training data improves model performance.


2. EgoQuestions is derived from Ego4D and HD-Epic. Several evaluated large models may have been pre-trained on data overlapping with Ego4D/ HD-Epic text or video metadata. The paper does not document whether evaluated models’ training/pretraining corpora could include these sources, nor does it provide a decontamination strategy.

3. The authors use a capable LLM to score answers, but do not report how well this automated scoring correlates with human judgments.

**Questions:**

1. Is EgoQuestions valuable beyond evaluation/diagnostics? Please provide training/fine-tuning experiments or clearly position the dataset as an evaluation/diagnostic resource only.

2. Were the evaluated large models pretrained on Ego4D or HD-EPIC, the sources from which EgoQuestions was drawn?


3. What is the agreement between the LLM judge and human annotators?

---

### Official Review · Reviewer_3o3B · 2025-10-28

**Soundness:** 1
**Presentation:** 1
**Contribution:** 2
**Rating:** 2
**Confidence:** 4

**Summary:**

This paper proposes an Egocentric Video QA benchmark. The authors say that many questions in current datasets don’t really test whether a model understands the video from the recorder’s point of view. Instead, models can often guess the answer using easy “shortcuts,” like recognizing the general scene or common actions. This gives a false impression that Vision-Language Models (VLMs) are better at first-person understanding than they actually are.

To fix this, the authors make three main contributions: Defining good egocentric questions, creating a question-generation pipeline, and building the EgoQuestions Benchmark. Using this pipeline, they produce EgoQuestions, a benchmark containing 2,500 carefully made question–answer pairs.

They then test several Vision-Language Models on both the new and existing datasets. The results show that current models perform much worse on the new, stricter questions. There’s about a 10% performance drop compared to the older, easier benchmarks. This confirms that existing benchmarks overestimate how well current models truly understand first-person videos.

**Strengths:**

- The paper provides insights about how to better build egocentric video understanding benchmarks by eliminating the shortcut in questions.

**Weaknesses:**

- Limited Question Scope: The benchmark does not explicitly assess more complex reasoning abilities, such as those involving long-term intentions, social dynamics, or the causal consequences of the recorder’s actions.

- Reliance on LLM-as-Judge: While using an LLM to evaluate open-ended responses is a common practice, it remains imperfect. The observed 10% $\Delta_{acc}$ performance gap might partly result from the judge model itself finding the F2 questions (which include shortcuts) easier to assess. Incorporating more reliable multiple-choice questions into the benchmark is recommended.

- Writing Quality: The paper’s writing requires improvement, as several key concepts—such as the SPO, SOAP, and SPAO subsets—are insufficiently defined.

- Insufficient Model Coverage: The number of evaluated models is too limited for a benchmark paper, which reduces the generality and robustness of the findings.

- Missing qualitative study: The qualitative analysis of benchmark data and wrong answers by different models is missing.

**Questions:**

- The definitions of the data subsets—“SPO (because we provide the Subject and Predicate in the question while inquiring about the Object), SOAP, and SPAO”—are unclear. The distinctions among these subsets are not well explained, and it is not evident why they differ. This section requires significant clarification and improvement in writing quality.

- Did you perform any human validation on a subset of the LLM-as-judge's scores? How can we be confident that the judge itself isn't systematically scoring the "flawed" F2 questions more leniently, thereby contributing to the inflated gap?

- The current question templates (SPO, SOAP, SPAO) are focused on grounding actions and objects. Do you believe your three principles and crafting pipeline could be extended to generate more complex, abstract egocentric questions, such as those reasoning about the recorder's intentions ("Why am I lifting the pan?") or beliefs?

- Could you provide significantly more qualitative examples of your benchmark data?

- How do the SOTA models fail on your T1 questions? Could you analyze some examples? For instance, when asked an SPO question ("What am I picking up?"), do models tend to name a plausible but incorrect object from the scene, hallucinate an object, or simply state they don't know?

---

### Official Review · Reviewer_cYqD · 2025-11-03

**Soundness:** 2
**Presentation:** 3
**Contribution:** 2
**Rating:** 4
**Confidence:** 4

**Summary:**

This paper critiques existing egocentric Video Question Answering (VideoQA) benchmarks, arguing that many of their questions are not genuinely "egocentric" and can be answered using simple scene or action recognition shortcuts. The authors propose three principles to define what constitutes a valid egocentric question: it must concern the video recorder, avoid obvious shortcuts, and can involve the recorder's intentions.
Based on these principles, the authors develop a complex, multi-stage pipeline that uses Large Language Models (LLMs) to parse narrations from existing video datasets (Ego4D, HD-Epic) into grammatical components. These components are then reassembled into new questions using code templates. The result is a new benchmark of 2,500 QA pairs, named EgoQuestions. The paper evaluates several Vision-Language Models (VLMs) on this new benchmark and on a set of "flawed" counterpart questions, demonstrating that model performance is substantially lower on their rigorously crafted questions. This suggests that previous benchmarks may overestimate the true egocentric reasoning capabilities of current models.

**Strengths:**

1. Important Problem Identification: The paper correctly identifies and thoroughly diagnoses a subtle but important flaw in existing egocentric VideoQA benchmarks. The analysis of how shortcuts can lead to an overestimation of model capabilities is a valuable contribution to the community.

2. Clear Conceptual Framework: The three principles proposed for defining an egocentric question are intuitive, well-reasoned, and provide a useful conceptual vocabulary for future benchmark design. This formalization is a key strength of the paper.

3. Analysis of Existing Benchmarks: The quantitative analysis in Table 1, showing that a very low percentage of questions in popular benchmarks adhere to their principles (e.g., ~10-11% for EgoTempo and EgoTextVQA), provides strong empirical evidence for the paper's central claims.

**Weaknesses:**

1. Limited Scope of Contribution: The paper is primarily a dataset paper. While dataset contributions are valuable, this one is accompanied by neither a novel model nor a new learning technique. Its main finding is that "harder questions are harder," which, while important to demonstrate, may not be seen as a sufficiently impactful research finding on its own.

2. The evaluation of open-ended answers relies on an LLM-as-a-judge. Could the authors provide a more extensive comparison between their LLM judge and human annotators (beyond a simple accuracy metric) to better understand its biases and reliability? For instance, what is the inter-rater reliability (e.g., Cohen's Kappa) between the LLM and human judges on a representative subset of the data? This would add much-needed credibility to the results in Table 3.

3. Given the multi-step LLM pipeline for parsing grammar and refining questions, what is the estimated end-to-end success rate of the generation process? How many initial narrations result in flawed or discarded questions due to cascading errors in the pipeline, and what measures are in place to ensure the final 2,500 questions are free of subtle artifacts introduced by this process?

4. Principle 2 aims to avoid shortcuts by, for example, not simultaneously naming an action and an object. This is handled by parsing grammar. However, shortcuts can be semantic rather than purely grammatical. How do the authors ensure their automated pipeline robustly distinguishes between a helpful contextual clue and an undesirable shortcut that a model could exploit?

**Questions:**

This paper has a strong motivation and a thoughtful conceptual framework for improving egocentric VideoQA. The diagnosis of the problem is excellent. However, the work is let down by significant methodological concerns in both its data generation and, most critically, its evaluation protocol. The reliance on an LLM-as-a-judge to score open-ended answers is a major weakness that undermines the paper's goal of bringing more rigor to the field.  Hope the authors could address the questions in the above sections.

---

### Author Response · Authors · 2025-11-13
**Thank you for the reviews**

We sincerely thank all the reviewers for the insightful feedback. We will carefully address your comments and improve the quality of our work accordingly.

We wish you all the best in your future research.

---

### Note · Authors · 2025-11-13

I have read and agree with the venue's withdrawal policy on behalf of myself and my co-authors.